# Anti-Oxidant Activity and Dust-Proof Effect of Chitosan with Different Molecular Weights

**DOI:** 10.3390/ijms20123085

**Published:** 2019-06-24

**Authors:** Yong Hyun Lee, So Yeon Park, Jae Eun Park, Byung Ok Jung, Jung Eun Park, Jae Kweon Park, You Jin Hwang

**Affiliations:** 1Department Biomedical Engineering, College of Health Science, 191 Hambakmoe-ro, Yeonsu-gu, Incheon 21936, Korea; yong9la2@naver.com; 2Department of Life Sciences, Gachon University, Seongnamdaero 1342, Seongnam-si, Gyeonggi-do 461-701, Korea; qkrth0413@naver.com; 3Department of Chemical and Biochemical Engineering, Dongguk University, 30, Pildong-ro 1-gil, Jung-gu, Seoul 04620, Korea; xenophiliusg@gmail.com; 4Institute of Red Snow Crab, Nonggongdanji-gil 76, Sokcho-si, Gangwon-do 24899, Korea; bok6738@hanmail.net; 5Department of Chemical and Biomolecular Engineering, Sogang University, 35 Baekbeom-ro, Mapo-gu, Seoul 04107, Korea; ireadl@naver.com

**Keywords:** chitosan, molecular weight, fenton reaction, oxidative cleavage, dust-proof

## Abstract

High molecular weight chitosan (HMWC) was degraded to prepare chitosan with different molecular weight based on the fenton reaction, which can produce aggressive OH-radicals produced from hydrogen peroxide in the presence of catalytic metal ions. The relative molecular weight, anti-oxidant activity, and fine dust removal effect of chitosan hydrolysates were elucidated to define their molecular weight and their potent biological activity. Our results demonstrate that chitosan hydrolysates derived from the hydrolysis of HMWC may possess significant free-radical scavenging activity as good anti-oxidants against the radical scavenging activity of DPPH and ABTS, respectively. Furthermore, chitosan hydrolysates can effectively eliminate fine dust, which may contain some particulate matter (PM) and unknown species of microorganisms from the air, suggesting that our data provide important information for producing air filters, dust-proof masks and skin cleaner for the purpose of human healthcare and well-being.

## 1. Introduction

Chitosan with different molecular weight was firstly considered as a potent factor for determining various biological activities such as anti-oxidant [1,2], anti-bacterial [3,4], anti-cancer [5,6], anti-fungal [7,8], etc. For several decades, many researchers have carried out studies to develop unique methods to prepare chitosan molecules with particular sizes. Therefore, several approaches, including chemical and enzymatic processes, have been applied to solve these problems for a long time. However, less is actually known about how these molecules can be prepared effectively and economically for use as potent bioactive materials. Since some bioactive macromolecules, like polysaccharides, can be degraded under oxidative reaction produced by reactive oxygen species (ROS) [9,10], we have tried to prepare chitosan hydrolysates with different molecular weights under these conditions. OH-radicals have been identified as specific species of ROS that are capable of degrading some polysaccharides isolated from natural resources [11,12].

A potent method known as the fenton reaction has been widely applied and is characterized as one of the most effective methods to produce OH-radicals in the presence of hydrogen peroxide (H_2_O_2_). As described in the previous study [13,14], this reaction results in oxidation of organic materials in the presence of H_2_O_2_, and some divalent metals ions such as Cu^2+^, Mg^2+^, and Zn^2+^. Herein, therefore, we focused on the degradation of high molecular weight chitosan (HMWC) using the fenton reaction to produce different sizes of molecular weight of chitosan, which can act as a potent bioactive agent for the elimination of anti-oxidants and fine dust, which can include some unknown air-borne microorganisms. Breathing of fine dust is considered the main cause of health problems such as skin diseases, lung cancer, and respiratory diseases.

Fine dust contains some unidentified particulate matter (PM), heavy metal ions, and, importantly, unknown microorganisms that are freely, widely and severely spreading through the whole nation. Depending on the size of the PM, various effects of fine dust can be observed in animals, fresh water, plants and human beings [15,16]. Therefore, it is necessary to define the contents of fine dust and develop a potent tool for protection from unexpected infection. Upon investigating the basic characteristics of some of the candidate molecules, a fine dust or particle proofing effect by using chitosan hydrolysates was further addressed in this study, including anti-oxidant activity and microbial growth inhibitory effect. Importantly, controlling the molecular weight of chitosan and elucidating those biological effects with respect to the requirements of various applications should be developed for human health. Regarding these concerns, the aim of this study is to elucidate the capability of fine dust proofing by using chitosan with different molecular weights.

## 2. Results and Discussion

### 2.1. Preparation of Chitosan Hydrolysates

To prepare biologically active substances, suitable degradation of various kinds of polysaccharides isolated from natural resources was performed and established in our previous study [17]. Upon investigating the significance of the fenton reaction for the degradation of several biomaterials, we currently tried to make chitosan with different molecular weights derived through the hydrolysis of high molecular weight chitosan (HMWC, ~600 kDa, 98.5% deacetylation degree, α-form) under optimal conditions. As described in the preliminary study, we found that the concentration of reducing end sugars derived through the hydrolysis of HMWC changed dramatically depending on the reaction conditions. As a matter of fact, however it is not the most suitable method for degradation of other polysaccharides, including fucoidan, alginate, and hyaluronic acids. On the other hand, it resulted in quite a rapid reaction for degrading HMWC into molecules of small size. Therefore, we focused on the production of the most active size of chitosan. As the result, we found the optimal hydrolysis of 1.0% HMWC was observed in the presence of 3% H_2_O_2_ to produce a high yield of the most active chitosan size. Also, we used 2.0 % HMWC to observe the different sizes of products under the same conditions. As a result, we obtained five samples that had been reacted for different lengths of time, and designated them as chitosan hydrolysates (CTSN) CTSN-P2, -P3, -B2, -B3, and -S3, respectively. Furthermore, chitin was also applied to the same reaction to elucidate the wide applicability of the fenton reaction as well. However, a significant amount of products was not produced by the hydrolysis of chitin. Based on these experiments, we suggest that these conditions for the fenton reaction might only be useful for chitosan.

Under the optimal reaction conditions, we found that CTSN-S3 had the highest concentration of reducing end sugars among the samples, as shown in Figure 1. This result suggests that decreasing the molecular weight of chitosan is dependent on reaction time and the concentration of H_2_O_2_ used as the constant chemical oxidative reagent. These hydrolysates were then applied to the subsequent experiments after the purification of chitosan hydrolysates had been performed using an immobilized metal affinity matrix (IMAM), since IMAM is also specialized at chelating with the most divalent metal ions [18,19], as described in the Materials and Methods.

### 2.2. Estimation of Molecular Weight of Chitosan Hydrolysates

Prior to determining the molecular weight of chitosan hydrolysates, the products were first visualized by TLC analysis to assess their differing size. However, we found that none of the hydrolysates were able to migrate to the upper part of the plate, instead staying at the bottom line. These results led us to propose that all products are much bigger than chito-oligosaccharides, consisting of more than 7 or 8 glucosamines linked together. We therefore strived to characterize the products that could possibly be bigger than the chito-oligosaccharides.

As described in our previous study, we demonstrated the biological usefulness of partially digested chitosan, referred to as medium molecular weight chitosan (MMWC), several times, including its antimicrobial activity [20,21]. Although we have been trying to develop the optimal conditions for preparing of MMWC or other specific sizes of products, it is very hard to control the reaction conditions through either chemical or enzymatic processes. Despite there being many reports describing the various biological or physiological activities of chitosan and its hydrolysates as potent biomedical or pharmaceutical agents, chitosan or its hydrolysates, consisting of several size of molecules, are still counted among the minor materials. Therefore, a single molecule that is bigger than heptameric glucosamine must be produced rapidly and efficiently in order to provide the scientific aspects of chitosan and its relative compounds in various research fields. As we determined based the final products by TLC analysis, the fenton reaction may provide a very rapid reaction for the degradation of HMWC, making it possible to produce molecules much bigger than heptameric glucosamine at high concentrations. Furthermore, it is obvious that reaction can be performed depending on the type of sugars, reaction time and concentration of H_2_O_2_ used the basic source of ROS production.

Following the TLC analysis, the molecular weight of each of the chitosan hydrolysates derived from the hydrolysis of HMWC by the fenton reaction was measured to provide more detailed information regarding the chitosan hydrolysates based on the values obtained from reducing the end sugar concentration and HPLC analysis, as described in the previous study [22,23]. As shown in Table 1, the molecular weights of the chitosan hydrolysates was estimated, ranging from 1.5 to 31 kDa, as we assumed that the products may be bigger than the chito-oligosaccharides consisting of 7 units of glucosamine. This result strictly correlated with the results of the TLC analysis. Taken together, we suggest that HWMC can rapidly be degraded by the combined reactions of metal ions in the presence of different concentrations of H_2_O_2_, suggesting the possibility of controlling the size through fine tuning the reaction conditions. This is a promising indication that the greatest honor will be accrued by the person who is able to produce a single molecule from the hydrolysis of HMWC in large quantities. This could open a much wider path for the characterization of the biological activities which have so far not been examined, and provide approaches for potent applications in various fields.

### 2.3. Measurement of Antioxidant Activity

To elucidate the bioactive function of chitosan hydrolysates, firstly, antioxidant activity was monitored. Prior to performing further assays, we hypothesized that the hydrolysates obtained after the fenton reaction may possess free radical sites on their structure. This means that all hydrolysates may be directly involved in the oxidant-reduction of the reaction. Obviously, a significant effect of the removal of the radical scavenging ability using the chitosan hydrolysates prepared in this study was observed against DPPH. However, there were no significantly different effects of those molecules between all tests for DPPH, ABTS, and FRAP assays, although 5.8 kDa (CTSN-B3) of hydrolysates showed relatively high activity toward ABTS, as shown in Table 2. These results are in agreement with other related results demonstrating the potent antioxidant activity of chitosan hydrolysates [24,25]. It has been suggested that differences in size or molecular weight might be potently involved in antioxidant effects. However, we found that there were no such big and significant differences as a result of the different molecular weights of chitosan in this study.

Based on this, it is not possible to conclude the real effect of chitosan with different molecular weights, because there have been no reports describing the purification and identification of pure chitosan hydrolysates as a single molecule that are bigger than 8 units of glucosamine. Therefore, our results still prompt us to consider the relationship between the molecular weight and biological functions of chitosan hydrolysates, prepared in several different types of reactions. Until the biological activities when using pure molecules have been characterized, discussion and reconsideration will continue. Nevertheless, we suggest that the chitosan hydrolysates prepared and tested in this study with molecular weights ranging from 1.5 to 31 kDa did not show any significant differences toward antioxidant activities.

### 2.4. Removal Effect of Fine Dust and Microorganisms

We further investigated a biological approach to assessing the potent bio-action of chitosan hydrolysates. For a couple of decades, we have been suffering from fine dust, which was unexpectedly imported by flying into our country from neighboring countries and/or produced by industrial plants in domestic areas. Therefore, the removal of fine dust using natural compounds that are less or non-toxic has recently been considered in order to develop a powerful tool or some particular agents to protect the breathing or rapidly remove them from our skin without suffering any allergic effects. We applied chitosan hydrolysates to address their effectiveness in the removal of fine dust and the microorganisms included in fine dust. As shown in Figure 2, fewer countable microorganisms were observed on the LB-agar solid medium compared to the control. However, this type of test, conducted by waiting for natural contamination by the air, may not be reproducible and is hard to control. Therefore, we further investigated the effects of microbial growth inhibition on a liquid culture system. As a result, we found that among the samples tested, CTSN-P2, -B2 and -S2 showed very effective activity in terms of microbial growth inhibition at a concentration of 200 µg/mL towards methicillin-resistant *Staphylococcus aureus* (MRSA) (Figure 3). Furthermore, microbial growth was significantly inhibited in the presence of 25 µg/mL chitosan hydrolysates compared with the control. Furthermore, we found that the exponential phase of MRSA was delayed for about 3~4 h compared with the control. What is the significance of a delay in lag time on microbial growth? With much smaller counts of microbes, these chitosan hydrolysates may have a good chance to attack or aggregate with microbes. In addition, this can provide time for an alternative treatment with other agents following treatment with chitosan hydrolysates in order to clean the skin through the use of masks and other means. Small changes can result in huge effects, like pricking a hole with a pin in a balloon filled with air or water. In this respect, the time-lagging delay of the microbial culture in the presence of chitosan hydrolysates should also be considered an important observation. Hereby, a question arises as to how it can we prove the full removing of copper ions derived during the fenton reaction. IMAM is how they can be commercialized and specialized for absorbing most divalent metal ions with high binding capacity, including Ni^2+^, Zn^2+^, Cu^2+^, Ca^2+^, Co^2+^ or Fe^2+^. Upon applying chitosan hydrolysates (CTSN) containing copper ions to the IMAM column, CTSN could be collected without significant loss in quantity based on the reducing sugar assay. In the preliminary experiment, we found that there were no detectable amounts of copper ions observed in the eluted CTSN samples under spectroscopic analysis using various concentrations of copper sulfate ranging from 0 to 100 ng/mL as standard at 780 nm. Subsequently, the control sample was prepared in the absence of HMWC and applied to the IMAM to assess whether any effect of the copper ions on the antimicrobial activity CTSNs was likely, as well. As a result, there was no significant effect of the control on bacterial growth. Therefore, taken together, we conclude that CTSN-P2, -B2 and -S2 could be used as a blocker for microbial species included in fine dust or from other contaminants, although CTSN-P2 showed a slightly greater effect than the others at the same concentration (Figure 3d).

In addition, we found that the effective removal of PM 2.5 or PM 10 in fine dust was observed after passing the air through the filter, which was either treated or non-treated with chitosan hydrolysates, compared with the control. These results obviously indicated that all chitosan hydrolysates may have highly potent affinity toward these PMs, as well. Among the chitosan hydrolysates, CTSN-P2 showed a higher removal efficiency of fine dust than the other chitosan hydrolysates and the control (Table 1 and Figure 4). Here, it is worth noting that it very difficult, based on this kind of study, to define real activity because of the effect of wind disturbing the air stream. Therefore, timing and the point at which it was performed were very important factors to control to obtaining reproducible observations. Although we do not know the real significance of the slight changes of values, our observation showed the very important message that chitosan may have a wide variety of actions, depending on its molecular weight, as demonstrated in this study and other earlier studies [25].

In addition, PMs from fine dust that had been captured on the surface of the mask were examined under the microscope. As shown in Figure 5, particles with different sizes and shapes were visualized as black dots with treatment with CTSN-B2, which was randomly selected for testing as it was the most plentiful sample among the chitosan hydrolysates. Since CTSN-B2 was applied to a single layer of the mask, it would be possible to enhance the removal efficiency of PMs, including unidentified microbes, from air contaminants when it is applied to a normal mask consisting of multiple layers of cotton. Commercialized masks are widely known to people as essential things to wear for dust proofing. However less is known about potent agents for both dust proofing and protection from microbial infection. Therefore, our observations are truly important to gaining a better understanding of the properties of fine dust, which leads to several diseases, as described in the introductory section.

## 3. Materials and Methods

### 3.1. Chemical and Materials

Chemicals for the determination of antioxidant activity such as L(+)-ascorbic acid (99.5%), iron sulfate heptahydrate, iron chloride (anhydrous, 98%), copper sulfate pentahydrate and 4-Hydroxy-benxhydrazid were purchased from Samchun Chemical Co. (Pyungtaek, Korea) or Sigma Chemical Co. (St. Louis, MO, USA), respectively. Hydrogen peroxide (35%), used for the fenton reaction, was purchased from Rackhee Pharmaceutical Co. (Paju, Korea). High molecular weight chitosan (HMWC) was kindly provided by the Sokchomulsan Co., Ltd. (Sokcho, Korea). The TLC plate used to analyze chitosan hydrolysates was purchased from Merck KGaA Co. (Darmstadt, Germany). All other chemicals used in the quantification and chemical reactions were first grade.

### 3.2. Preparation of Chitosan Hydrolysates

HWMC (2 g) was dissolved in 2% acetic acid (100 mL) to make 2% (*w*/*v*) of chitosan solution at 37 °C for 2 h. To bring the chitosan solution to the fenton reaction, H_2_O_2_, as a source for hydroxyl-radicals, was added to the solution so that the final concentration was 0~7.0%. The fenton reaction was performed in the presence or absence of H_2_O_2_ used as a control. Briefly, a mixture (20 mL) consisting of chitosan, H_2_O_2_, and copper sulfate (0.01%, *w/v*) was prepared and heated on a heat block at 80 °C for 10 min to 2 h to accelerate the degradation of HMWC. After the reaction finished, reactants were transferred onto ice to cool down the temperature. To separate undigested HMWC, samples were collected by centrifugation at 13,000 rpm for 5 min.

### 3.3. Purification of Chitosan Hydrolysates

To remove the contents of copper ions from the hydrolysates, freeze-dried hydrolysates were dissolved in 2% acetic acid to make 2% chitosan hydrolysate solutions. These solutions were passed through a column packed with immobilized metal affinity matrix (IMAM, Merck Millipore Corporation, Darmstadt, Germany), which was pre-equilibrated with 2% acetic acid in advance before loading the samples. Copper ions can tightly bind to the IMAM, and free chitosan hydrolysates in 2% acetic acid were passed through without standing inside the column. The chitosan hydrolysates in the solution were subsequently transferred to fresh tubes, and 1 M NaOH solution was added to neutralize the eluents. Other samples were prepared as well after rinsing the IMAM with 1 mM EDTA (pH 8.0) and distilled water using 2 bed volumes of the column, respectively. Precipitates were further separated by centrifugation at 13,000 rpm for 5 min, and rinsed with distilled water three times. All samples were then freeze-dried to make hydrolysates in powder form, and these were used for further study.

### 3.4. Quantification of Chitosan Hydrolysates

The efficiency of hydrolysis by the fenton reaction was determined by reducing sugar contents using PHABAH (4-Hydroxy-benxhydrazid) solution consisting of 0.025g PHABAH dissolved in 10 mL NaOH (0.5 mM), as described in the previous study [26]. The degree of the hydrolysis of HMWC by the fenton reaction under various conditions was simply monitored by the colorimetric method using a UV-spectrophotometer at a wavelength of 405 nm (Infinite M200 Pro Nanoquant, TECAN, Untersbergstrasse, Austria). The relative concentration of reducing sugar derived from the hydrolysis of HMWC was then determined by comparing it to the standardized values when prepared using glucosamine ranging from 0 to 1 mM. Furthermore, for the **c**hitosan hydrolysates derived from the hydrolysis of HMWC after the fenton reaction, 20 µL of each sample was loaded on the TLC plate to assign the hydrolysis pattern, as described in the previous study [27]. Finally, the end products for each condition were visualized after the TLC plate was heated at 180 °C, resulting in samples being detected as dark blots after providing sufficient time for the appearance of sugars. The molecular weight of each hydrolysate was measured based on the values obtained in the previous study by following the correlations established with reducing sugar assay and HPLC analysis [28].

### 3.5. Antioxidant Activity of Chitosan Hydrolysates

To investigate the effect of molecular weight difference, the antioxidant activity of chitosan hydrolysates derived from the hydrolysis of HMWC by the fenton reaction was determined by the 1,1-diphenyl-2-picrylhydrazyl (DPPH) radical scavenging assay, 2,2′-azino-bis-3-ethylbenzthiazoline-6-sulphonic acid (ABTS) radical scavenging assay and Ferric Reducing Antioxidant Power (FRAP) assay, respectively, as described in the previous study [22]. Ascorbic acid (5 mg/mL) was used as a positive control. The antioxidant activity of chitosan hydrolysates with different molecular weights was determined to calculate the relative activity and specific activity to evaluate the potential significance of chitosan hydrolysates as biological materials.

### 3.6. Growth Inhibitory Effect of Chitosan Hydrolysates on Microorganisms

Samples in the presence or absence of chitosan hydrolysates with different molecular weights were prepared in 0.1% in 0.1% acetic acid, respectively. These solutions were then spread onto a solid agar-LB medium plate, the lid was opened, and the solutions were air-dried and let stand for 5 min ouside the door. After that, the lids were closed and these plates were stored at 37 and 25 °C for 1–2 days for culturing bacterial and fungal species. The growth inhibitory effect of chitosan hydrolysates on the unidentified microorganisms included in find dust was then monitored and recorded by photograph.

### 3.7. Removal Effect of Find Dust

A simple hand-made tester was located at outside door to collect fine dust and other air contaminants. Simplified filters with or without spraying of chitosan hydrolysates were set on a box, sucked air to the inside of the box using a vacuum system (9–10 volt) for 30 min, and collected air contaminants using a portable machine called Air Quality Monitor (BR-SMART 126, Bramc, Trivex, Singapore). The concentration of particulate matter (PM) with different sizes in fine dust was measured and recorded in a real-time monitoring system, following the manufacturer’s instructions. The removal effect of fine dust or air contaminants by chitosan hydrolysates was also monitored by using a microscope (ECLIPSE TS100, Nikon, Tokyo, Japan), and then calculated by comparing it to the values of the control.

For the statistical analysis, the average values and standard deviations (± S.D.) were calculated from at least three different experiments. The significance of the values was determined by applying Student’s *t*-test (*p* < 0.05).

## 4. Conclusions

Chitosan hydrolysates with different molecular weights were prepared in a rapid reaction, referred to as fenton oxidative cleavage, in the presence of catalytic metals. Unexpectedly, we found that there was no significant antioxidant activity toward the testing assays. However, a couple of hydrolysates, referred to as CTSN-P2, -B2 or -S2, showed significant removal effect of fine dust and the microorganisms included in air-borne fine dust. Taken together, our data suggest that hydrolysates of chitosan can block or form tight complexes with the components in fine dust, or microbial species in the air contaminants. Although it is not currently possible to explain the exact mechanism for how these effectively work, taken together, we conclude that chitosan hydrolysates with different molecular weights ranging from 1.5 to 31 kDa can be used as potent agents blocking or making tight complexes with microbial species and fine dust in the air.

## Figures and Tables

**Figure 1 ijms-20-03085-f001:**
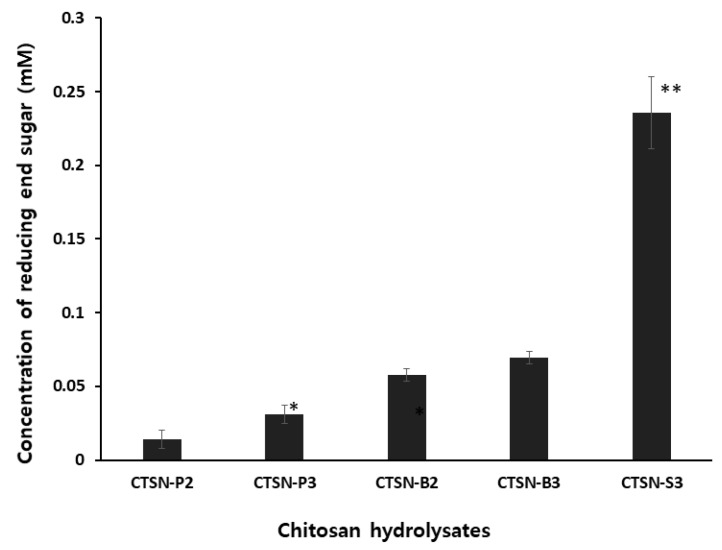
Determination of the concentration of reducing end sugars. Chitosan hydrolysates designated as CTSN-P, -B and -S were derived from the hydrolysis of high molecular weight chitosan by the fenton reaction. CTSN-P2, -P3, -B2, -B3, and -S3 series were obtained after reaction times of 5, 10, 15, 20 and 30 min, respectively. The values obtained from three separate experiments are presented as averages with standard deviation (± S.D.). * *p* < 0.05, ** *p* > 0.05.

**Figure 2 ijms-20-03085-f002:**
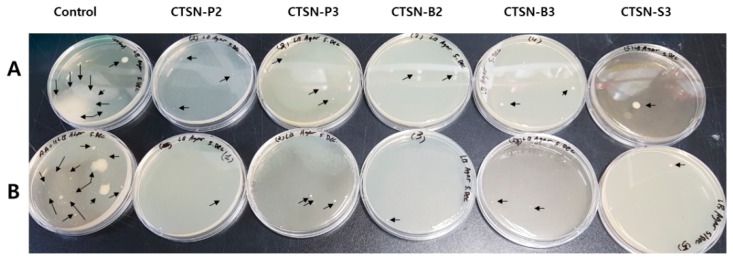
Effect of chitosan hydrolysates on bacterial growth. Agar-solid mediums for fungus (**A**) and bacteria (**B**) were placed in an open area for 5 min, covered, and cultured at room temperature and 37 °C overnight or for up to 3 days, respectively. Microorganisms included in the fine dust were counted and are indicated by arrows.

**Figure 3 ijms-20-03085-f003:**
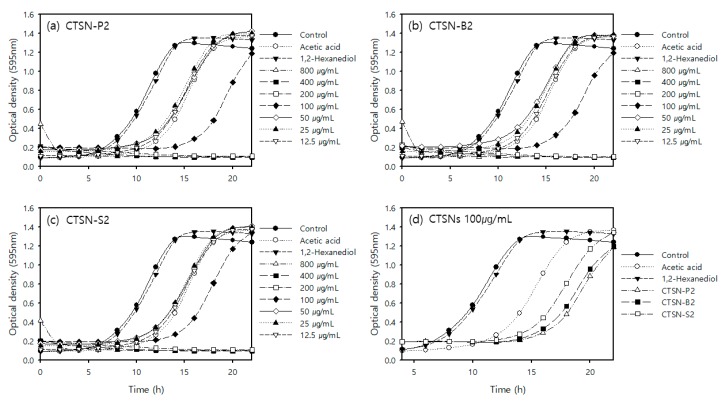
Concentration dependence of chitosan hydrolysates on bacterial growth. MRSA was cultured in LB liquid medium at 37 °C overnight in the presence or absence of chitosan hydrolysates ranging from 0 to 800 µg/mL. Chitosan hydrolysates included: (**a**) CTSN-P2, (**b**) CTSN-B2, (**c**) CTSN-S2. (**d**) The growth inhibitory effect of chitosan hydrolysates was monitored at the same concentration (100 µg/mL) to define the effective molecular weight of chitosan. Acetic acid (0.1%, *v*/*v*) was used as a control for the base solvent of chitosan, and 1,2-Hexandiol (0.1%, *v/v*) was used as a commercialized agent for antimicrobial chemicals. Growth of microorganisms collected on the surface of the mask was monitored by optical density at 595 nm in a time-dependent manner up to 24 h.

**Figure 4 ijms-20-03085-f004:**
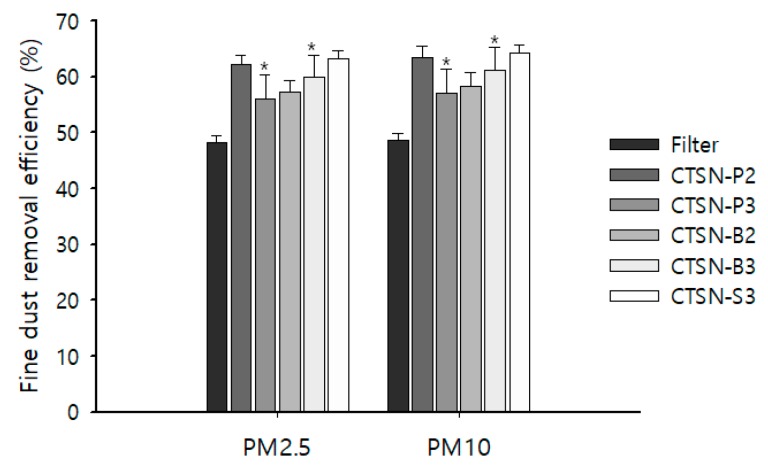
Fine dust removal efficiency of chitosan hydrolysates. The fine dust adsorption efficiency was measured using an Air Quality Monitor, taking the outside air passed the through filter to the inside of the chamber. The air quality was monitored based on the adsorption efficiency of find dusts by chitosan hydrolysates CTSN-P2, -P3, -B2, -B3, and -S3 toward PM 2.5 and PM 10, respectively. Values were obtained from at least three separate experiments and presented as averages with standard deviations (mean ± SD (*n* = 3). * *p* > 0.05).

**Figure 5 ijms-20-03085-f005:**
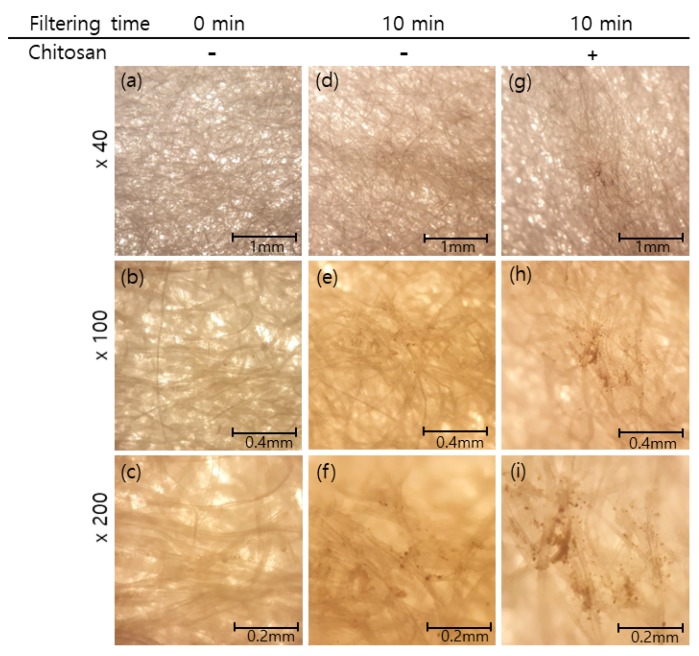
Dust-proofing effects of filter membrane treated with chitosan hydrolysates CTSN-B2. The filter surfaces were observed by microscopic analysis (at magnifications of ×40, ×100 and ×200). Observations are as follows: (**a**–**c**) the filter surface without treatment (time zero) as control; (**d**–**f**) the filter surface after passing air into the chamber for 10 min; (**g**–**i**) the filter surface treated with CTSN-B2 after passing air into the chamber for 10 min.

**Table 1 ijms-20-03085-t001:** Antioxidant activity of chitosans with different molecular weights.

		CTSN-P2	CTSN-P3	CTSN-B2	CTSN-B3	CTSN-S3
Relative activity (%) *	DPPH	90.78	91.27	90.81	91.59	91.07
	FRAP	N/D	N/D	N/D	N/D	N/D
	ABTS	17.41	21.86	17.75	24.78	20.09
Av MW (kDa) **		31	30	8.1	5.8	1.5

* Values are indicated as the average of three separate experiments. ** Av MW (kDa): average molecular weight of chitosan hydrolysates derived from the hydrolysis of high molecular weight chitosan.

**Table 2 ijms-20-03085-t002:** Fine dust removal efficiency of chitosan-treated filter.

Filter Type	Av MW (kDa) *	Fine Dust Removal Efficiency (%)	
		PM 2.5	PM 10
Filter (Control)	-	48.29 ± 1.41	48.68 ± 1.49
Filter + CTSN-P2	31	62.25 ± 1.92	63.49 ± 2.45
Filter + CTSN-P3	30	56.08 ± 5.14 **	57.00 ± 5.26 **
Filter + CTSN-B2	8.1	57.18 ± 2.62	58.27 ± 3.07
Filter + CTSN-B3	5.8	59.92 ± 4.78 **	61.10 ± 5.18 *
Filter + CTSN-S3	1.5	63.13 ± 1.82	64.24 ± 1.65

* Av MW: average molecular weight (kDa). ** *p* > 0.05.

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
