# Peer review of "Anti-Oxidant Activity and Dust-Proof Effect of Chitosan with Different Molecular Weights"

_ijms, 2019, doi:10.3390/ijms20123085_

Reviewer 1 Report

This manuscript presents the preparation and the potential applications of chitosan hydrolysates. Relative molecular weight, antioxidant activity, and fine dust removal effect of the degraded chitosans are discussed in the aim to define the molecular weight and their potent biological activity.

The presented data are of interest for the scientific community involved in chitosan derivatives applications as biomaterials.

Author Response

Comments by Reviewer #1

General comments

1. This manuscript presents the preparation and the potential applications of chitosan hydrolysates. Relative molecular weight, antioxidant activity, and fine dust removal effect of the degraded chitosans are discussed in the aim to define the molecular weight and their potent biological activity. The presented data are of interest for the scientific community involved in chitosan derivatives applications as biomaterials.

Response #1.

Thank you for your careful consideration

Reviewer 2 Report

The paper should be reconstructed: the methods and materials sections should be replaced between 1. Introduction and 2. Result and Discussion for better reading.

Abbreviations should be in descending order full names, such as CTSN appears at the first time on page 2, verse 34, without revealing the full name. An identity, the HMWC abbreviation appears in the Introduction section without specifying a full name. Although this abbreviation is defined in Abstract, it also suggests in the Introduction section to provide a full name.

Authors should also characterize the initial chitosan properties subjected to the degradation process (degree of the deacetylation, molecular weight, crystallinity). They should also provide evidence of the absence of the presence of Cu in tested hydrolysates so as to confirm the results of the microbiological tests obtained.

Author Response

Comments by Reviewer #2

Comments

Comment #1. the methods and materials sections should be replaced between 1. Introduction and 2. Result and Discussion for better reading.

Response #1. Thank you for pointing out that the editing order is wrong. We have modified it in order. 1. Introduction, 2. Methods and materials, 3. Results and Discussion, 4. Conclusion

Comment #2. Abbreviations should be in descending order full names, such as CTSN appears at the first time on page 2, verse 34, without revealing the full name.

â–¶ Page 3, line 20-21

Response #2 : We added the full name of DPPH, ABTS and FRAP, 1,1-diphenyl-2-picrylhydrazyl (DPPH) radical scavenging assay, 2,2′-azino-bis-3-ethylbenzthiazoline-6-sulphonic acid (ABTS) radical scavenging assay and Ferric Reducing Antioxidant Power (FRAP) assay,

â–¶ Page 4, line 17

Response #2 : We added the full name of CTSN. (CTSN : Chitosan hydrolysates)

Comment #2. An identity, the HMWC abbreviation appears in the Introduction section without specifying a full name. Although this abbreviation is defined in Abstract, it also suggests in the Introduction section to provide a full name.

â–¶ Page 2, line 11

Response #2. We added the full name of HMWC. (HMWC : high molecular weight chitosan)

Comment #3. Authors should also characterize the initial chitosan properties subjected to the degradation process (degree of the deacetylation, molecular weight, crystallinity).

â–¶ Page 2, line 11

Response #3. We added the initial chitosan properties in Result and Discussion section page 4, line 7. [high molecular weight chitosan (HMWC, ~600 kDa, 98.5% deacetylation degree, α-form)]

Comment #4. They should also provide evidence of the absence of the presence of Cu in tested hydrolysates so as to confirm the results of the microbiological tests obtained.

â–¶ Page 6, line 42 ~ Page 7, line 10

Response #4. We explained the fenton reaction in Result and Discussion section page 6, line 42 ~ Page 7, line 10. The following is added.

[Herein a question rises to ask how can we prove the full removing of copper ions derived during the fenton reaction. IMAM is commercialized and specialized for absorbing most of divalent metal ions such as Ni2+, Zn2+, Cu2+, Ca2+, Co2+ or Fe2+ with high binding capacity. Upon applying chitosan hydrolysates (CTSN) containing copper ions on to the IMAM column, CTSN can be collected without significant loss in quantity based on reducing sugar assay. In the preliminary experiment, we found that there was no detectable amounts of copper ions observed in the eluted CTSN samples under spectroscopic analysis using various concentration of copper sulfate ranging from 0 to 100 ng/mL as standard at 780 nm. Subsequently, control sample was prepared in the absence of HMWC and applied to the IMAM to assess if there is any effect of copper ions on antimicrobial activity likely CTSNs as well. As the result, there was no any significant effect of the control on bacterial growth (data not shown). ]